# Health effects of occupational noise exposure on heavy-duty equipment operators and exposed workers in a mining firm in Ghana

Francis Amanle Cudjoe[1,2], Douglas Aninng Opoku[1,3]*, Nana Kwame Ayisi-Boateng[3,4], Joseph Osarfo[5], Kofi Sekyere Boateng[6], Lydon Nii Adjiri Sackey[6], Edgar Andoh Cobbina[1,2], Isaac Kofi Yankson[7], Alhassan Sulemana[6]

1 Department of Occupational and Environmental Health, School of Public Health, Kwame Nkrumah University of Science and Technology, Kumasi, Ghana, 2 Gold Fields Ghana Limited, Tarkwa Mine, Tarkwa, Ghana, 3 University Hospital, Kwame Nkrumah University of Science and Technology, Kumasi, Ghana, 4 Department of Medicine, School of Medicine and Dentistry, Kwame Nkrumah University of Science and Technology, Kumasi, Ghana, 5 Department of Community Health, School of Medicine, University of Health and Allied Health Sciences, Ho, Ghana, 6 Department of Environmental Science, College of Science, Kwame Nkrumah University of Science and Technology, Kumasi, Ghana, 7 Council for Scientific and Industrial Research-Building and Road Research Institute, Kumasi, Ghana

* douglasopokuaninng@gmail.com

## Abstract

### Background

The mining industry is one of the sectors that uses heavy-duty equipment in its daily operations. This exposes miners to undesirable noise levels, increasing their risks of health-related problems. However, published data on the health effects of occupational noise exposure on miners in Ghana are limited, and this can affect potential interventions to promote miners' health and safety. This study, therefore, assessed noise-exposure levels and associated health-related problems among heavy-duty equipment operators and other exposed workers in a mining firm in Ghana.

### Methods

A cross-sectional study involving 316 randomly selected heavy-duty equipment operators and exposed workers was conducted from 29th March 2023–31st May 2023. Data on socio-demographic and work-related characteristics, including age, mining experience, knowledge of noise-induced hearing loss (NIHL), noise exposure levels and health-related problems, were collected using a pretested questionnaire. Logistic regression models were used to identify significant predictors of health-related problems.

### Results

The mean age of study participants was 33.8 (±7.5) years with a range of 21–60 years. The prevalence of health-related problems in the twelve months before the

**Data availability statement:** All relevant data are within the manuscript and its Supporting Information files.

**Funding:** The author(s) received no specific funding for this work.

**Competing interests:** The authors have declared that no competing interests exist.

study was 55.7%. The commonly reported health-related problems included hearing difficulties (84.1%), hearing loss (49.4%), and sleeping difficulties (36.9%). Approximately 68.6% of the workers were exposed to noise levels that were unacceptable. After adjusting for significant covariates, factors such as working experience of 5–9 years (AOR: 4.25, 95%CI: 1.92–9.40), inadequate knowledge of NIHL (AOR:1.78, 95%CI: 1.03–3.09) and exposure to unacceptable noise levels (AOR = 5.52, 95%CI = 2.91–10.48) were independently associated with health-related problems.

## Conclusion

The prevalence of health-related problems among the workers was high. Potential strategies, including a hearing conservation program to promote health and safety among these workers at the workplace, should target reducing the exposure to high noise levels and increasing awareness of NIHL.

## Introduction

Noise pollution is considered one of the significant occupational hazards in the mining sector. It is a major issue of concern to public health and safety due to its adverse effects on both the environment and human health [1]. Globally, noise pollution is on the ascendancy due to the introduction and application of sophisticated machines at the workplace to reduce the physical burden of workload on employees, and also to increase the rate of productivity [2–4].

The adverse health effects of mining on miners cannot be ignored, despite the sector's contributions to economic growth. The nature of mining activities comes along with several health externalities [5,6]. The mining sector is one of the occupations that uses heavy-duty equipment and machines in its daily operations, and thus, predisposes employees to various hazards and risks of occupation-related health problems [7]. The International Labour Organization (ILO) reports that almost three million workers, including miners worldwide, lose their lives annually due to occupation-related illnesses, injuries and accidents, a rise of more than 5.0% from 2015 [8]. A study in Ontario, Canada, found that among occupational sectors, workers in the mining and construction sector had the highest relative risk of work-related fatalities compared to those in the trade, services and manufacturing sectors [7].

Some of the rigorous activities in the mining sector that expose miners to high noise levels include excavation, hauling of materials, dumping, earth-moving, material handling, and road construction. The exposure of miners to excessive noise could negatively affect both their psychological and physical well-being [9,10]. The prolonged exposure of miners to noise levels above 75 dB can result in adverse health problems, including cardiovascular diseases, hearing impairment, and sleep disturbances [11–15]. Also, communication may be affected when working in a noisy environment, and this can result in workplace accidents and injuries [16]. The Occupational Safety and Health Administration (OSHA) recommends strategies including

hearing conservative programs for workers whose activities expose them to noise levels of 85 dBA and above for more than eight hours, to enhance their health and safety [17].

In Ghana, previous studies have reported several health externalities, including respiratory diseases [6], hearing impairments [18], and occupational injuries among gold miners [19]. However, despite evidence of a significant effect of exposure to occupational noise on health-related problems, there is limited published data on the subject in Ghana among heavy-duty equipment operators in the mining sector, an occupation known to expose workers to high noise levels at the workplace. Previous studies on this subject were limited to quarry workers in the Kwabre East District in the Ashanti Region [20], whilst a similar study conducted in the southwestern part of Ghana [21] failed to assess the potential effect of noise exposure and prior knowledge of NIHL on the possible occurrence of health-related problems, resulting in a knowledge gap that the current study aims to address. The current study was therefore conducted to assess the health-related problems and associated factors, including possible effects of occupational noise exposure and knowledge of occupational NIHL among heavy-duty equipment operators and exposed workers in a mining firm in Ghana. This study provides essential data to inform policies aimed at ensuring that heavy-duty equipment operators and exposed workers are not subjected to undesirable noise in the mining sector, a crucial step in promoting their health and safety.

## Methods

### Study design and setting

An analytical cross-sectional study was conducted at a mining firm in the Western Region of Ghana from 29th March 2023, to 31st May 2023.

The Western Region is one of Ghana's sixteen administrative regions and is known for its timber, gold, and bauxite resources. Due to the region's favourable geology, concessions for gold mining activities abound, in addition to its being the home of some of the largest and most established mining companies in the country. The mining firm where the current study was conducted has been granted a mining lease by the Ghana Minerals Commission to operate within a land area of 20,825 ha. The firm mainly engages in surface mining, and it has over 2600 workers with different training and expertise. Of the number of workers in this firm, 730 are heavy-duty equipment operators. The mining firm has one hundred and eighty-two heavy-duty equipment operators. The primary operations of heavy-duty equipment operators involve loading and hauling run-of-mine ore to the crushing plant, as well as waste materials to designated dumping sites.

### Study population, sample size determination and sampling technique

The study population consisted of heavy-duty equipment operators and exposed workers at the mining firm. The exposed workers were those whose activities subjected them to noise exposure from the heavy-duty equipment. This heavy-duty equipment included dump trucks, backhoes, bulldozers, excavators, graders and loaders. All heavy-duty equipment operators who had been engaged for at least 12 months before the data collection were included in the study. Heavy-duty equipment operators with a history of hypertension and occupation-related illnesses before their engagement in the current work were excluded from the study. This information was derived through a self-report.

The sample size was determined using the Charan and Biswas formula [22] . Using a 95% confidence interval (1.96) (z), an error margin of 5.0% (E), and 24.0% of mine workers who experienced NIHL (P) in a previous study [23], a sample size of 280 was obtained. With a non-response rate of 10.0%, a minimum sample size of 308 was arrived at for this study.

The selection of study participants was done through a simple random sampling approach. The staff list was retrieved from the human resource department and entered into a ballot using the lottery method. All staff names were de-identified by assigning codes to them. The codes were written on a piece of paper, folded and placed in a bowl. To ensure that the folded papers were evenly mixed, the bowl was shaken with an independent person drawing the papers one after the

other until the estimated sample size was obtained. All the codes were traced to the corresponding names and contacted to schedule an appointment for interviews.

## Data collection tools and techniques

A structured questionnaire (see S1 File) was used for data collection. The questionnaire was uploaded to a smartphone using a Google form and administered to study participants during face-to-face interviews. The questionnaire was made up of four sections, namely socio-demographic and work-related characteristics (age, sex, level of education, exposure to whole-body vibrations, mining experience, etc.), knowledge of occupational NIHL, exposure to noise levels and health-related problems. Mining experience in this study was operationalized as the number of years of work in the mining firm. Three postgraduate students were engaged and trained for three days on administering questionnaires, consent process, and translating the questionnaire into Twi (a local Ghanaian language) by the principal investigator, who has a background in occupational health and safety. This was done to enhance the consistency in the data collection. The questionnaire was administered in both English and Twi (for participants who opted for it) languages to the study participants. Prior to that, the questionnaire was pretested among thirty heavy-duty equipment operators and exposed workers in a mining firm in the Ashanti Region of Ghana with relevant changes made to it before using it to collect the final study data. The pretest was conducted to improve the internal validity and reliability of the data collection instrument.

## Measures

### Derivation of the outcome variable

The outcome variable for this study was a health-related problem in the 12 months before the data collection. This was derived by asking study participants if they had experienced any health-related problem in the 12 months before the data collection, with a binary response of "yes" and "no" and coded as "1" for "yes" and "0" for "no". Those who reported experiencing health-related problems were asked to tick more than one disease (from a list) where applicable and mention them, if not provided on the list. Sleep difficulty in this study was defined as having trouble falling asleep or staying asleep at night.

### Derivation of independent variables

**Knowledge of occupational NIHL.** Participants' knowledge of occupational NIHL was measured using ten (10) questions on the causes, effects, and prevention of NIHL. A correct response to a question was assigned a score of '1', and a wrong response was given a score of '0'. The composite score was computed and expressed as percentages, with participants categorized as either having adequate knowledge (score of ≥50%) or inadequate knowledge (score of <50%).

**Occupational noise exposure levels.** The noise exposure levels of study participants were retrieved from their health records. Specifically, their last average noise exposure assessment conducted in the twelve months before the study. Noise monitoring in the mining firm was conducted by an occupational hygienist using a calibrated sound level meter. The sound level meter was positioned 30 centimetres around the hearing zone of the operator, and three consecutive readings were taken five minutes apart, and an average was recorded. All those who had an average exposure limit of more than 85 dBA were categorized as having an unacceptable exposure limit, while those who had ≤ 85 dBA were categorized as having an acceptable exposure limit based on the World Health Organization's (WHO) recommendation [24].

## Data quality management and statistical analysis

The quality of the data was enhanced by checking for completeness and accuracy by a data validation team. The data was downloaded from the Google form in Microsoft Excel, cleaned and exported to Stata version 16 for analysis.

Descriptive statistics were performed and presented using frequencies, percentages, means, standard deviations, medians, interquartile ranges and graphs. The noise exposure levels among heavy-duty equipment operators and exposed workers by the department were compared using one-way ANOVA. The Bivariate and multivariate logistic regression analyses were performed to identify independent factors associated with health-related problems. A bivariate logistic regression analysis was used to determine the relationship between health-related problems and independent variables, including socio-demographic and work-related factors, exposure to occupational noise and knowledge of occupational NIHL. In the multivariable analysis, independent factors that showed a significant association ($p < 0.05$) with health-related problems in the bivariate logistic regression analysis were adjusted for and presented as adjusted odds ratio with 95.0% confidence interval. All independent significant associations were maintained at p-value <0.05.

### Ethical considerations

Ethical approval was obtained from the Committee on Human Research, Publications and Ethics, School of Medicine and Dentistry, Kwame Nkrumah University of Sciences and Technology, Kumasi (reference number: CHRPE/AP/189/23). Also, written approval to conduct the study at the mining company was obtained from their management. All the study participants provided both verbal and signed or thumb-printed written informed consent form. The aim of the study was explained to all the study participants and they were made to understand that their participation was entirely voluntary. Hence, they had the right to reject participation or withdraw from the study even after accepting to participate without having to suffer any penalty or their welfare affected at the company.

## Results

### Demographic and work-related characteristics of study participants

A total of 316 participants were recruited for this study. The mean age of study participants was 33.8 [±7.5) years, with a range of 21–60 years. Majority (84.2%) of the workers were males, and approximately 45.6% of them had tertiary education. Over half (54.4%) of the workers were married. The median work experience of the workers was 4 (interquartile range: 2, 7) years with a range of 1–25 years and over 48.4% of them were at the mining department. Over two-thirds (68.7%) of the workers worked for 6 days a week, and approximately 56.0% of them were exposed to whole-body vibration for more than two hours per day at the workplace (Table 1).

### Health-related problems among study participants

Generally, over half (55.7%) of the workers reported having experienced health-related problems in the twelve months before the study. Among the health-related problems reported by the workers included hearing difficulties (84.1%), hearing loss (49.4%), sleeping difficulties (36.9%), hypertension (19.3%), ear infections (21.0%), and ringing in the ears [10.2%] [(Table 2).

### Noise exposure levels among study participants

The median noise exposure among the workers was 87.0 (interquartile range: 81.0, 89.7) dB with a minimum exposure level of 38 dB and a maximum exposure level of 96 dB. About 31.4% of the workers were exposed to acceptable levels of noise in the past six months, while the remaining 68.6% (219) of them were exposed to unacceptable levels of noise (Fig 1).

Workers in the mining department were exposed to the highest level of noise in the past twelve months (mean exposure levels = 87.0 dB, SD = ±6.8 dB) with the lowest mean exposure levels observed among those in the health, safety and environment department (mean exposure levels = 74.4 dB, SD = ±15.1 dB) (Table 3).

### Knowledge of occupational NIHL among study participants

Generally, over half (55.7%) of the workers who were recruited as study participants had adequate knowledge about NIHL. About 93.4% of the workers were aware of NIHL. Over 99.4% (314) of the study participants indicated exposure to

**Table 1. Socio-demographic and work-related characteristics of study participants.**

| Characteristics | Frequency, N = 316 | Percentage, % (Range) |
|---|---|---|
| **Age group years)** | | |
| <30 | 98 | 31.0 |
| 30–39 | 143 | 45.3 |
| 40+ | 75 | 23.7 |
| Mean age (±SD) | 33.8 (±7.5) | (21–60) |
| **Sex** | | |
| Male | 266 | 84.2 |
| Female | 50 | 15.8 |
| **Level of education** | | |
| No formal education | 58 | 18.4 |
| Basic | 36 | 11.4 |
| Secondary | 78 | 24.7 |
| Tertiary | 144 | 45.6 |
| **Religion** | | |
| Christian | 268 | 84.8 |
| Muslim | 48 | 15.2 |
| **Marital status** | | |
| Single | 144 | 45.6 |
| Married | 172 | 54.4 |
| **Mining experience (years)** | | |
| <5 | 180 | 57.0 |
| 5–9 | 82 | 26.0 |
| 10+ | 54 | 17.0 |
| Median working experience (IQR) | 4 (IQR: 7, 2) | (1–25) |
| **Work department** | | |
| Engineering | 88 | 27.8 |
| HSE | 26 | 8.2 |
| Geology | 20 | 6.3 |
| Metallurgical | 29 | 9.2 |
| Mining | 153 | 48.4 |
| **Work days in a week (days)** | | |
| 5 | 60 | 19.0 |
| 6 | 217 | 68.7 |
| 7 | 39 | 12.3 |
| **Availability of restroom** | | |
| Yes | 100 | 31.6 |
| No | 216 | 68.4 |
| **Exposure to whole-body vibrations** | | |
| Never | 4 | 1.3 |
| Occasionally | 83 | 26.3 |
| <2hrs per day | 52 | 16.5 |
| >2hrs per day | 177 | 56.0 |
| **Usage of HPD during work** | | |
| Yes | 309 | 97.8 |
| No | 7 | 2.2 |
| **Frequency of wearing HPD (n = 308)** | | |

*(Continued)*

**Table 1.** (Continued)

| Characteristics | Frequency, N=316 | Percentage, % (Range) |
|---|---|---|
| Always | 253 | 80.1 |
| Sometimes | 55 | 17.9 |

IQR, interquartile range; SD, standard deviation; HPD, hearing protection device.

**Table 2.** Health-related problems among study participants.

| Characteristics | Frequency, N=316 | Percentage, % |
|---|---|---|
| **Experienced health-related problems** | | |
| Yes | 176 | 55.7 |
| No | 140 | 44.3 |
| **Health-related problems experienced (n=176)*** | | |
| Hearing difficulties | 148 | 84.1 |
| Diagnosed with hearing loss in the past 12 months | 87 | 49.4 |
| Sleeping difficulty | 65 | 36.9 |
| Hypertension | 34 | 19.3 |
| Ear infections | 37 | 21.0 |
| Ringing in the ears | 18 | 10.2 |

*Multiple responses.

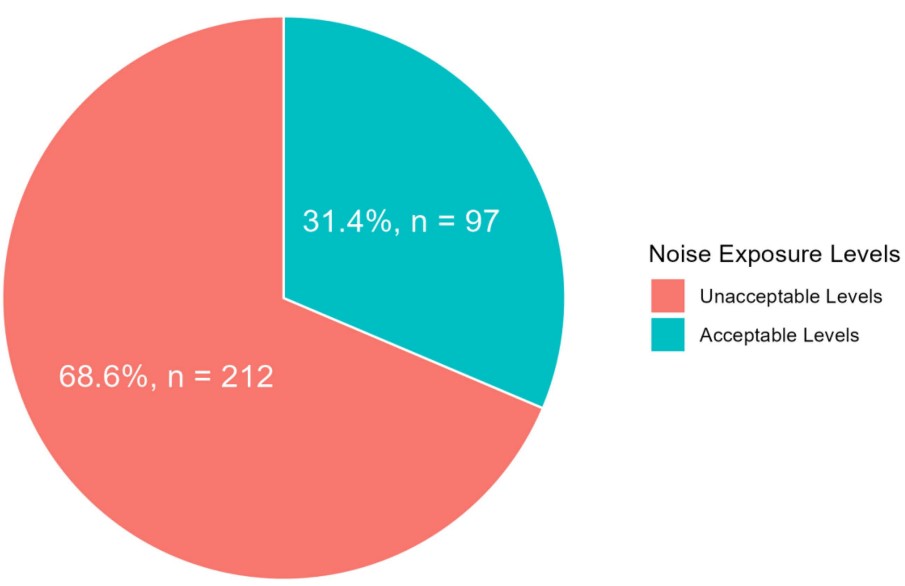

**Fig 1.** Noise exposure levels.

**Table 3. Noise exposure levels of heavy-duty equipment operators by department.**

| Variables | Mean (dB) | Standard deviation (dB) |
|---|---|---|
| Mining | 87.0 | ±6.8 |
| Geology | 86.8 | ±5.0 |
| Metallurgy | 84.6 | ±10.2 |
| Engineering | 81.8 | ±6.8 |
| HSE | 74.4 | ±15.1 |

NB: p<0.001.

excessive noise can result in NIHL. Over half (50.6%, 160) of the workers indicated that NIHL can be corrected medically and surgically. Over 98.7% of the workers indicated that NIHL can cause workplace injuries and accidents. Approximately 58.5% of the workers reported that the use of PPE can help reduce NIHL. About 97.8% of the workers indicated that a person with NIHL can have a problem with hearing high-pitched sounds. Approximately 55.1% (174) of the workers reported that a person with NIHL will have trouble understanding conversations when he/she is in a noisy place (Table 4).

## Factors associated with health-related problems among study participants

After adjusting for significant independent variables, factors such as mining experience, knowledge of occupational NIHL and exposure to occupational noise were identified as independent predictors of health-related problems.

Workers with mining experience of 5–9 years and more than 10 years had about four times (AOR: 4.25, 95%CI: 1.92–9.40) and (AOR: 4.46, 95%CI: 1.60–12.38), respectively increased odds of experiencing health-related problems compared to those with less than five years of mining experience. Workers who had inadequate knowledge of NIHL had about 78% (AOR:1.78, 95%CI: 1.03–3.09) increased odds of experiencing health-related problems compared to those who had adequate knowledge of NIHL. Workers who were exposed to unacceptable levels of noise had about six times (AOR = 5.52, 95%CI = 2.91–10.48) increased odds of experiencing health-related problems compared to those who were exposed to acceptable levels of noise (Table 5).

## Discussions

The significance of gold mining to a country's economy cannot be overemphasized. Nevertheless, it is important to consider the general cost and adverse health externalities associated with workers' exposure to occupational hazards from mining activities. One way to reduce the risks of exposure to occupational hazards is by setting up preventive strategies, including administrative and engineering controls or the use of personal protective equipment (PPE). To improve miners' health and safety, it is important to identify all the potential factors that could increase their risk of work-related illnesses, diseases, and accidents and introduce the appropriate interventions to address them.

Exposure to noise at the workplace is of significant concern, particularly in the mining sector, given its adverse health effects on workers. The current study found that most (68.6%) of the workers were exposed to noise levels above the WHO's recommended minimum threshold of 85dB(A) [24]. This finding resonates with earlier reports that the mining sector is prone to noise pollution [25,26]. The noise exposure levels observed in the present study are at variance with a study in China, which reported that about 31.9% of non-coal miners were exposed to unacceptable levels of noise at the workplace [27]. The observed differences could be attributed to variations in the study population, the tools used for measuring noise, and advancements in technology in mining. Among the workers exposed to undesirable levels of noise, those in the geology and mining departments were exposed to noise levels that exceeded the minimum threshold of 85 dB(A) recommended by the WHO. The nature of their (workers in geology and mining departments) activities involved blasting and excavation, which exposed these workers to undesirable noise. This implies that exposure to occupational

**Table 4. Knowledge of noise-induced hearing loss among study participants.**

| Characteristics | Frequency, N=316 | Percentage, % |
|---|---|---|
| **Level of knowledge** | | |
| Adequate | 176 | 55.7 |
| Inadequate | 140 | 44.3 |
| **Awareness of NIHL** | | |
| Yes | 295 | 93.4 |
| No | 21 | 6.6 |
| **Exposure to excessive noise can result in NIHL** | | |
| Yes | 314 | 99.4 |
| No | 2 | 0.6 |
| **NIHL can be corrected medically and surgically** | | |
| Yes | 160 | 50.6 |
| No | 156 | 49.4 |
| **NIHL can cause workplace injuries and accidents** | | |
| Yes | 312 | 98.7 |
| No | 4 | 1.3 |
| **The following can help reduce NIHL** | | |
| Getting closer to the source of the noise | 2 | 0.6 |
| Prolonged exposure to loud noise | 2 | 0.6 |
| Frequently taking breaks to limit the duration of exposure to loud noise | 26 | 8.2 |
| Using the appropriate PPE | 185 | 58.5 |
| Developing a good equipment maintenance culture | 101 | 32.0 |
| **WHO recommended noise exposure limit** | | |
| 70dB over 24 hours and 85dB over 1 hour | 20 | 6.3 |
| 60dB over 24 hours and 85dB over 2 hours | 18 | 5.7 |
| 85dB over 24 hours and 85dB over 3 hours | 278 | 88.0 |
| **A person with NIHL can have a problem with hearing high-pitched sounds** | | |
| Yes | 309 | 97.8 |
| No | 7 | 2.2 |
| **A person with NIHL will not have trouble understanding conversations when he/she is in a noisy place** | | |
| Yes | 142 | 44.9 |
| No | 174 | 55.1 |
| **Degeneration of inner ear structures occurs over time and is associated with NIHL** | | |
| Yes | 309 | 97.8 |
| No | 7 | 2.2 |
| **A symptom of NIHL does not include asking others to speak more slowly and clearly** | | |
| Yes | 140 | 44.3 |
| No | 176 | 55.7 |
| **A person with mild NIHL may hear some speech sounds but soft sounds are hard to hear** | | |
| Yes | 306 | 96.8 |
| No | 10 | 3.2 |

PPE, Personal protective equipment.

**Table 5. Influence of noise exposure on the health of study participants.**

| Characteristics | Unadjusted OR (95%CI) | P-value | Adjusted OR (95%CI) | P-value |
|---|---|---|---|---|
| **Age group (years)** | | | | |
| <30 | 1.00 | | 1.00 | |
| 30–39 | 1.71 (1.01–2.87) | 0.044 | 1.28 (0.65–2.50) | 0.477 |
| 40+ | 1.60 (0.87–2.94) | 0.127 | 0.52 (0.19–1.37) | 0.185 |
| **Sex** | | | | |
| Male | 1.00 | | 1.00 | |
| Female | 0.43 (0.23–0.79) | 0.007 | 0.69 (0.30–1.61) | 0.394 |
| **Marital status** | | | | |
| Single | 1.00 | | – | – |
| Married | 1.01 (0.65–1.58) | | – | – |
| **Level of education** | | | | |
| No formal education | 1.00 | | | |
| Basic | 0.79 (0.34–1.82) | 0.579 | – | – |
| Secondary | 0.91 (0.46–1.82) | 0.797 | – | – |
| Tertiary | 0.86 (0.46–1.59) | 0.626 | – | – |
| **Mining experience (years)** | | | | |
| <5 | 1.00 | | 1.00 | |
| 5–9 | 3.21 (1.82–5.64) | <0.001 | 4.25 (1.92–9.40) | <0.001 |
| 10+ | 2.72 (1.43–5.19) | 0.002 | 4.46 (1.60–12.38) | 0.004 |
| **Work department** | | | | |
| Engineering | 1.00 | | | |
| HSE | 0.46 (0.12–1.21) | 0.117 | – | – |
| Geology | 1.26 (0.48–3.32) | 0.645 | – | – |
| Metallurgical | 1.35 (0.58–3.12) | 0.488 | – | – |
| Mining | 2.75 (1.60–4.72) | <0.001 | – | – |
| **Work days in a week (days)** | | | | |
| 5 | 1.00 | | 1.00 | |
| 6 | 3.64 (1.95–6.78) | <0.001 | 1.30 (0.57–3.01) | 0.534 |
| 7 | 9.80 (3.76–25.57) | <0.001 | 3.23 (0.99–10.57) | 0.053 |
| **Level of knowledge** | | | | |
| Adequate | 1.00 | | 1.00 | |
| Inadequate | 1.79 (1.13–2.81) | 0.012 | 1.78 (1.03–3.09) | 0.039 |
| **Noise exposure level** | | | | |
| Acceptable | 1.00 | | 1.00 | |
| Unacceptable | 6.81 (3.96–11.71) | <0.001 | 5.52 (2.91–10.48) | <0.001 |

OR, odds ratio; CI, confidence interval.

noise varied across departments. Hence, interventions such as a hearing conservation program to reduce noise exposurein this population must be department-specific.

The prevalence of health-related problems in this study was alarming. The results indicated that more than half (55.7%) of the workers experienced health-related problems in the twelve months before the study. This finding corroborates earlier reports that miners are at an increased risk of occupational health-related problems, including illnesses, injuries and accidents [19,21]. Of the health-related problems experienced, the most commonly reported included hearing-related

difficulties and sleep difficulties. Hearing-related difficulties, being the predominant health-related problem in the current study, could be explained by most workers' exposure to unacceptable levels of noise and whole-body vibrations at the workplace. This finding aligns with previous studies, which reported a higher prevalence of hearing loss among workers exposed to both whole-body vibrations and noise compared to those who were not exposed [28,29].

The current study observed high adherence to hearing protection devices (HPD) among the study participants. The use of HPD by most of the workers is particularly encouraging, considering that it can offer protection to their ears by reducing the intensity of noise entering the wearer's eardrum. Additionally, adherence to the wearing of HDP is beneficial preventive measure for minimizing the occurrence of health-related problems, including NIHL [30,31]. It is therefore essential that training targeted at enhancing the effective use of the HPD is promoted at all costs by the mining firm to achieve one hundred per cent adherence due to its significance in promoting employees' health and safety.

The current study revealed that the number of years of mining was significantly associated with experiencing health-related problems, consistent with studies conducted among noise-exposed workers in Ethiopia [32], and China [33,34]. It was observed from the present study that mining experience of 5 years or more was associated with increased odds of experiencing a health-related problem. A possible reason could be that miners with 5 years or more of mining experience may have been exposed to high noise for prolonged periods exceeding the exposure duration limit, thereby predisposing them to health-related problems. In Ethiopia, a study found that metal workers who had been exposed to high noise levels with 6 years or more had about three to five times increased odds of experiencing noise-induced prehypertension compared to those with 1–5 years of working experience [32].

The association between exposure to high noise and the development of a health-related problem have been well documented in the literature [28,32–35]. The current study revealed that exposure to unacceptable noise levels was associated with about six times increased odds of experiencing a health-related problem. The WHO recommends that workers are not continually exposed to noise levels exceeding 85 dB (A) for more than an hour due to its adverse health effects on workers [15]. Exposure to undesirable/unacceptable noise levels can result in sleep disturbance, hypertension, noise-induced hearing loss and adverse effects on both the cardiovascular and metabolic systems [15,32,34]. We recommend that, aside from strict adherence to wearing HPD, the mining firms should also encourage workers exposed to high noise to take frequent breaks or rotate schedules.

In this study, four out of ten of the miners had inadequate knowledge of NIHL. Inadequate knowledge of NIHL in the current study was associated with a 78.0% increased odds of experiencing a health-related problem. A possible reason could be that workers with inadequate knowledge of NIHL may not be aware of or well-informed about the adverse effects of exposure to high noise on their health and hence, may not adhere strictly to preventive measures, including the use of hearing protectors and taking frequent breaks to reduce the duration of exposure. A study in Jordan reported that hearing information about NIHL resulted in about 56.3% willingness to use earplugs [36]. We recommend that the management of mining firms frequently organize training programs aimed at increasing the workers' awareness of NIHL.

## Limitations of the study

The use of a self-report approach in measuring health-related problems in the current study is a limitation. Again, the use of a cross-sectional design limits our ability to draw causal inferences between the factors associated with health-related problems. Also, the study was carried out in only one mining company in Ghana, which limits its generalizability.

## Conclusion

This study found that most of the study participants experienced health-related problems in the twelve months before the study, which was influenced by factors such as inadequate knowledge of NIHL, mining experience of 5 years or more and exposure to unacceptable noise levels at the workplace. We recommend that regulators, such as the Labour and Minerals Commissions of Ghana should increase monitoring and enforcement of regulatory standards at mining companies to

ensure that workers are not exposed to unacceptable levels of workplace noise. Also, we recommend that the mining firm implement interventions such as organizing workshops to improve workers' knowledge of NIHL, and encourage workers who work in departments whose activities expose them to high noise levels to take more frequent breaks, thereby limiting their duration of noise exposure. Future studies can adopt a longitudinal study design to evaluate the effects of existing interventions on reducing exposure to workplace noise and their impact on the occurrence of health-related problems.

## Supporting information

**S1 File. Questionnaire for study data.**
(DOCX)

**S2 File. Datasets used for the analysis.**
(XLS)

## Acknowledgments

We are grateful to the management of the mining company for permitting us to conduct this study. We also appreciate all the heavy equipment operators who took part in this study.

## Author contributions

**Conceptualization:** Francis Amanle Cudjoe, Douglas Aninng Opoku, Nana Kwame Ayisi-Boateng, Alhassan Sulemana.

**Data curation:** Francis Amanle Cudjoe, Douglas Aninng Opoku, Joseph Osarfo, Kofi Sekyere Boateng, Lydon Nii Adjiri Sackey, Edgar Andoh Cobbina, Isaac Kofi Yankson.

**Formal analysis:** Douglas Aninng Opoku.

**Methodology:** Francis Amanle Cudjoe, Douglas Aninng Opoku, Nana Kwame Ayisi-Boateng, Joseph Osarfo, Lydon Nii Adjiri Sackey, Isaac Kofi Yankson, Alhassan Sulemana.

**Project administration:** Francis Amanle Cudjoe, Kofi Sekyere Boateng, Edgar Andoh Cobbina, Isaac Kofi Yankson, Alhassan Sulemana.

**Supervision:** Alhassan Sulemana.

**Validation:** Francis Amanle Cudjoe, Douglas Aninng Opoku, Nana Kwame Ayisi-Boateng, Joseph Osarfo, Kofi Sekyere Boateng, Lydon Nii Adjiri Sackey.

**Writing – original draft:** Francis Amanle Cudjoe, Douglas Aninng Opoku, Alhassan Sulemana.

**Writing – review & editing:** Francis Amanle Cudjoe, Douglas Aninng Opoku, Nana Kwame Ayisi-Boateng, Joseph Osarfo, Kofi Sekyere Boateng, Lydon Nii Adjiri Sackey, Edgar Andoh Cobbina, Isaac Kofi Yankson, Alhassan Sulemana.

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
