## [Decision Letter · Decision Letter 0]

23 Jun 2024

Dear Dr. Opoku,

Thank you for submitting your manuscript to PLOS ONE. After careful consideration, we feel that it has merit but does not fully meet PLOS ONE’s publication criteria as it currently stands. Therefore, we invite you to submit a revised version of the manuscript that addresses the points raised during the review process.

We look forward to receiving your revised manuscript.

Kind regards,

Stephen Dajaan Dubik, BSc, MPH, MPhil

Academic Editor

PLOS ONE

Journal Requirements:

Reviewers' comments:

Reviewer's Responses to Questions

**Comments to the Author**

1. Is the manuscript technically sound, and do the data support the conclusions?

Reviewer #1: Yes

Reviewer #2: Yes

2. Has the statistical analysis been performed appropriately and rigorously?

Reviewer #1: Yes

Reviewer #2: Yes

3. Have the authors made all data underlying the findings in their manuscript fully available?

Reviewer #1: Yes

Reviewer #2: Yes

4. Is the manuscript presented in an intelligible fashion and written in standard English?

Reviewer #1: Yes

Reviewer #2: Yes

Reviewer #1: PONE-D-24-02308

The authors have investigated the association between exposure to noise and experiencing work-related health outcomes. This study is relevant to the current state of the mining sector in Ghana, and I highly recommend it be published. The manuscript is well written however I recommend that the authors consider the following and refine the manuscript before publication.

1. There is an inconsistency noted within the manuscript concerning the participants of the study. While it is frequently indicated that the respondents were operators of heavy-duty machinery, Table 1 enumerates individuals from departments such as HSE, which do not engage in the operation of such equipment. The authors must clarify and reconcile this discrepancy.

2. I would suggest that the authors include the questionnaire used in their study as a supplementary file. This would enhance the transparency and reproducibility of the research.

3. A minor textual amendment is necessary on Page 5, Line 128, where the term "operators" should be inserted after "heavy-duty equipment" to maintain clarity in the description of the subjects.

Reviewer #2: I would like to thank the authors for such an important study. This is an essential area of research the authors seek to address.

Abstract.

Conclusion: The conclusion of the abstract must be reviewed. The conclusion should be directed to the study population. It is not all miners who were studied in this research.

Background.

The background is fairly written well.

Line 92. The word "close" should be revised.

Methods

Line 103: The authors seek to describe the study setting. It is not clear whether the authors are referring to a firm or a number of firms. They should consider revising the write up from line 103 to 112 to rightly describe the study setting.

The sentence beginning in line 120 "These workers were excluded ......." should be deleted.

Outcome variable: It is not too clear what health-related problems were used to define the outcome variable. Are they related to the outcome of interest?

Results:

What is the definition for Mining experience? Does it relate to years of work as an heavy-equipment operator or what?

What does "ear surgery" means? Is it after a health-related problem has occurred and surgery preformed or is a disease name?

Discussion:

The discussion is lacking some details point that will demonstrate the merit of the study. Paragraph 2 should be revised, more can be said that has policy implications. PPE was not discussed. Does heavy-equipment operators use PPEs? This was not discussed as it is important in the mining industry.

**Do you want your identity to be public for this peer review?** For information about this choice, including consent withdrawal, please see our Privacy Policy

Reviewer #1: No

Reviewer #2: No

---

## [Author Response · Author response to Decision Letter 1]

20 Aug 2024

Reviewer #1

The authors have investigated the association between exposure to noise and experiencing work-related health outcomes. This study is relevant to the current state of the mining sector in Ghana, and I highly recommend it be published. The manuscript is well written however I recommend that the authors consider the following and refine the manuscript before publication.

Reviewer’s comment: There is an inconsistency noted within the manuscript concerning the participants of the study. While it is frequently indicated that the respondents were operators of heavy-duty machinery, Table 1 enumerates individuals from departments such as HSE, which do not engage in the operation of such equipment. The authors must clarify and reconcile this discrepancy.

Authors’ response: The study population were heavy-duty equipment operators and other exposed workers. We included other exposed workers because their activities also exposed them to the noise from the heavy-duty equipment. We have provided clarity in the revised manuscript and have also used it consistently throughout the manuscript. The revised section now reads;

The study population was heavy-duty equipment operators and exposed workers in the mining firm. The exposed workers were those whose activities subjected them to noise exposure from the heavy-duty equipment.

Reviewer’s comment: I would suggest that the authors include the questionnaire used in their study as a supplementary file. This would enhance the transparency and reproducibility of the research.

Authors’ response: The questionnaire has been included as a supplementary file as recommended by the reviewer. The section in reference now reads;

A structured questionnaire (see Supplementary File 1) was used for data collection

Reviewer’s comment: A minor textual amendment is necessary on Page 5, Line 128, where the term "operators" should be inserted after "heavy-duty equipment" to maintain clarity in the description of the subjects.

Authors’ response: We are grateful to the reviewer for drawing our attention to these errors. This has been done as suggested and it now reads as;

The selection of study participants was done through a simple random sampling approach.

Reviewer #2:

Reviewer’s comment: I would like to thank the authors for such an important study. This is an essential area of research the authors seek to address.

Authors’ response: We appreciate the reviewer’s time and effort in ensuring that the quality of the manuscript is enhanced.

Reviewer’s comment: Conclusion: The conclusion of the abstract must be reviewed. The conclusion should be directed to the study population. It is not all miners who were studied in this research.

Authors’ response: The conclusion has been revised. It now reads;

The prevalence of health-related problems among the workers was high. Working experience, knowledge of NIHL and exposure to unacceptable noise levels were significant factors. Potential strategies including a hearing conservation program to promote health and safety among these workers at the workplace should target reducing the exposure to high noise levels and increasing awareness of NIHL.

Reviewer’s comment: The background is fairly written well. Line 92. The word "close" should be revised.

Authors’ response: The word ‘close’ has been revised as recommended by the reviewer. The revised statement now reads as;

………… to assess the possible effect of noise exposure and prior knowledge of NIHL on the possible occurrence of health-related problems, resulting in a knowledge gap that the current study aims to explore.

Reviewer’s comment: Line 103: The authors seek to describe the study setting. It is not clear whether the authors are referring to a firm or a number of firms. They should consider revising the write up from line 103 to 112 to rightly describe the study setting.

Authors’ response: We are grateful to the reviewer for this insightful comment. The study was conducted in a mining firm in the Western Region of Ghana. Clarity has been provided in the revised manuscript to show that the work was conducted in only one firm and not ‘firms’.

Reviewer’s comment: The sentence beginning in line 120 "These workers were excluded ......." should be deleted.

Authors’ response: The sentence in reference has been deleted from the revised manuscript as recommended by the reviewer.

Reviewer’s comment: Outcome variable: It is not too clear what health-related problems were used to define the outcome variable. Are they related to the outcome of interest?

Authors’ response: Yes, the outcome of interest in this study was experiencing a health-related problem in the last twelve months before the study. Clarity has been provided in the revised manuscript.

Reviewer’s comment: What is the definition for Mining experience? Does it relate to years of work as an heavy-equipment operator or what?

Authors’ response: Mining experience in this study was operationalized as the number of years of work in the mining firm. This has been defined in the revised manuscript under “Data collection tools and techniques”.

Reviewer’s comment: What does "ear surgery" means? Is it after a health-related problem has occurred and surgery preformed or is a disease name?

Authors’ response: We have dropped the variable ‘‘ear surgery’’ from the revised manuscript.

Reviewer’s comment: The discussion is lacking some details point that will demonstrate the merit of the study. Paragraph 2 should be revised; more can be said that has policy implications. PPE was not discussed.

Authors’ response: The discussion has been revised including paragraph 2. PPE (hearing protection device) has also been discussed as recommended by the reviewer.

Reviewer’s comment: Does heavy-equipment operators use PPEs? This was not discussed as it is important in the mining industry.

Authors’ response: Yes they use hearing protection device and this has been discussed in the revised manuscript.

The current study observed high adherence to hearing protection device (HPD) among the study participants. The use of HPD by most of the workers is particularly encouraging considering that it can offer protection to their ears by reducing the intensity of exposed noise entering the eardrum at the workplace. Additionally, adherence to HDP is a very useful preventive measure for minimizing the occurrence of health-related problems, including NIHL (30,31). It is therefore important that educational workshops targeted at enhancing the use of HPD is promoted at all cost due to its significance in promoting employees’ health and safety at the workplace.

---

## [Decision Letter · Decision Letter 1]

2 Sep 2025

Health effects of occupational noise exposure on heavy-duty equipment operators and exposed workers in a mining firm in Ghana

PONE-D-24-02308R1

Dear Dr. Opoku,

We’re pleased to inform you that your manuscript has been judged scientifically suitable for publication and will be formally accepted for publication once it meets all outstanding technical requirements.

Kind regards,

Vidya Ramkumar, Ph.D

Academic Editor

PLOS ONE

Additional Editor Comments (optional):

Reviewer #2:

Reviewers' comments:

Reviewer's Responses to Questions

**Comments to the Author**

Reviewer #2: All comments have been addressed

2. Is the manuscript technically sound, and do the data support the conclusions?

Reviewer #2: Yes

3. Has the statistical analysis been performed appropriately and rigorously?

Reviewer #2: Yes

4. Have the authors made all data underlying the findings in their manuscript fully available?

Reviewer #2: Yes

5. Is the manuscript presented in an intelligible fashion and written in standard English?

Reviewer #2: Yes

Reviewer #2: The authors have addressed all my previous comments. They mentioned the study's limitations but failed to discuss its strengths. Besides, I don't have any more comments.

**Do you want your identity to be public for this peer review?** For information about this choice, including consent withdrawal, please see our Privacy Policy

Reviewer #2: No

---

## [Editor Report · Acceptance letter]

PONE-D-24-02308R1

PLOS ONE

Dear Dr. Opoku,

I'm pleased to inform you that your manuscript has been deemed suitable for publication in PLOS ONE. Congratulations! Your manuscript is now being handed over to our production team.

Kind regards,

on behalf of

Dr. Vidya Ramkumar

Academic Editor

PLOS ONE